# Intra and Inter-Device Reliabilities of the Instrumented Timed-Up and Go Test Using Smartphones in Young Adult Population

**DOI:** 10.3390/s24092918

**Published:** 2024-05-03

**Authors:** Thâmela Thaís Santos dos Santos, Amélia Pasqual Marques, Luis Carlos Pereira Monteiro, Enzo Gabriel da Rocha Santos, Gustavo Henrique Lima Pinto, Anderson Belgamo, Anselmo de Athayde Costa e Silva, André dos Santos Cabral, Szymon Kuliś, Jan Gajewski, Givago Silva Souza, Tacyla Jesus da Silva, Wesley Thyago Alves da Costa, Railson Cruz Salomão, Bianca Callegari

**Affiliations:** 1Laboratório de Estudos da Motricidade Humana, Av. Generalíssimo Deodoro 01, Belém 66073-000, PA, Brazil; thamelathais.terapeuta@gmail.com; 2Department of Physiotherapy, Speech Therapy and Occupational Therapy, Faculty of Medicine, University of São Paulo, São Paulo 05403-000, SP, Brazil; pasqual@usp.br; 3Instituto de Ciências Biológicas, Universidade Federal do Pará, Belém 66075-110, PA, Brazil; luis.monteiro@icb.ufpa.br (L.C.P.M.); givagosouza@ufpa.br (G.S.S.); 4Instituto de Ciências Exatas e Naturais, Universidade Federal do Pará, R. Augusto Corrêa, 01, Belém 66093-020, PA, Brazil; enzogabrielrocha29@gmail.com (E.G.d.R.S.); gpinto@ufpa.br (G.H.L.P.); 5Instituto Federal de São Paulo, Piracicaba 17607-220, SP, Brazil; abelgamo@gmail.com; 6Programa de Pós Graduação em Ciências do Movimento, Universidade Federal do Pará, Av. Generalíssimo Deodoro 01, Belém 66073-000, PA, Brazil; anselmocs@ufpa.br; 7Centro de Ciências Biológicas e da Saúde, Universidade do Estado do Pará, Tv. Perebebuí, 2623-Marco, Belém 66087-662, PA, Brazil; andre.cabral@uepa.br; 8Faculty of Rehabilitation, Józef Piłsudski University of Physical Education in Warsaw, Marymoncka 34, 00-968 Warsaw, Poland; szymon.kulis@awf.edu.pl; 9Faculty of Physical Education, Józef Piłsudski University of Physical Education in Warsaw, Marymoncka 34, 00-968 Warsaw, Poland; jan.gajewski@awf.edu.pl; 10Núcleo de Medicina Tropical, Universidade Federal do Pará, Rua Augusto Corrêa 01, Belém 66075-110, PA, Brazil; 11Centro de Ciências Biológicas e da Saúde-Campus VIII, Universidade Estadual do Pará, Av. Helía, s/n-Amapá, Marabá 68502-100, PA, Brazil; tacylasilva4@gmail.com (T.J.d.S.); wesley.tadcosta@aluno.uepa.br (W.T.A.d.C.); railsoncruzsalomao@gmail.com (R.C.S.)

**Keywords:** biomechanical phenomena, smartphone, test reproducibility, program evaluation, research instrument evaluation, timed-up and go test

## Abstract

The Timed-Up and Go (TUG) test is widely utilized by healthcare professionals for assessing fall risk and mobility due to its practicality. Currently, test results are based solely on execution time, but integrating technological devices into the test can provide additional information to enhance result accuracy. This study aimed to assess the reliability of smartphone-based instrumented TUG (iTUG) parameters. We conducted evaluations of intra- and inter-device reliabilities, hypothesizing that iTUG parameters would be replicable across all experiments. A total of 30 individuals participated in Experiment A to assess intra-device reliability, while Experiment B involved 15 individuals to evaluate inter-device reliability. The smartphone was securely attached to participants’ bodies at the lumbar spine level between the L3 and L5 vertebrae. In Experiment A, subjects performed the TUG test three times using the same device, with a 5 min interval between each trial. Experiment B required participants to perform three trials using different devices, with the same time interval between trials. Comparing stopwatch and smartphone measurements in Experiment A, no significant differences in test duration were found between the two devices. A perfect correlation and Bland–Altman analysis indicated good agreement between devices. Intra-device reliability analysis in Experiment A revealed significant reliability in nine out of eleven variables, with four variables showing excellent reliability and five showing moderate to high reliability. In Experiment B, inter-device reliability was observed among different smartphone devices, with nine out of eleven variables demonstrating significant reliability. Notable differences were found in angular velocity peak at the first and second turns between specific devices, emphasizing the importance of considering device variations in inertial measurements. Hence, smartphone inertial sensors present a valid, applicable, and feasible alternative for TUG assessment.

## 1. Introduction

The Timed-Up and Go (TUG) test is widely used by healthcare professionals as a means to evaluate various aspects of mobility [1]. Aligned with the International Classification of Functioning, Disability and Health, the TUG test serves as a quick and straightforward tool, highlighting the importance of gait and transfers in the performance of daily activities [2]. In addition to its role in the assessment of fall risk, the TUG test is also used to evaluate the functionality of the lower extremities and overall mobility [3]. 

Currently, the clinical interpretation of the TUG test revolves around a single parameter, that is, the time taken by individuals to complete the test [2]. It is conventionally established that a duration exceeding 15 s indicates a high risk of falls [3]. Despite the growing application of the TUG test across various populations, uncertainties persist about its accuracy in predicting fall risk outcomes in specific groups [2].

To address these concerns and facilitate a more objective evaluation of individuals’ performance, technological devices have been introduced in the assessment of the TUG test [4]. These devices not only provide additional data for complementary analyses but also have the potential to automate the test, allowing its administration in a home environment [5]. These advancements represent significant progress in the field of functional assessment, which benefits various populations.

Accelerometers and gyroscopes are currently the most commonly used sensors to instrument the standard TUG test, in a method known as instrumented TUG (iTUG) [5]. Extensive research has focused on parameters such as test duration, average joint angles, maximum trunk angle change, number of steps, threshold, and total foot contact time, as evidenced by a systematic review conducted by Ponciano et al. (2020) [3]. Using built-in inertial sensors, smartphones have emerged as a particularly promising tool for health assessment, including the evaluation of the iTUG test. In particular, the widespread use of smartphones in daily life eliminates the need for specific equipment acquisition, making them a convenient option. However, it is imperative to validate and assess the reliability of smartphone inertial sensors to fully understand their diagnostic potential [3].

Table 1 shows numerous studies that have demonstrated the applicability of inertial sensors in smartphones for iTUG execution (see Table 1). The scarcity of validation against gold standards and reliability studies remains evident. Gold standard validation in the context of assessing a measurement tool or diagnostic test involves comparing its results to those obtained from a widely accepted, definitive, or established reference method or criterion. In the case of the Timed-Up and Go (TUG) test and its instrumented version (iTUG), the gold standard could involve direct observation of the participant’s performance by trained healthcare professionals (i.e., with a chronometer measurement) and detailed motion analysis using sophisticated laboratory equipment that provides accurate and reliable measurements of mobility, gait, and fall risk (i.e., motion camera acquisition). Furthermore, reliability studies are crucial in assessing the consistency and dependability of measurement tools or diagnostic tests, especially in healthcare contexts. In the context of the TUG test and iTUG, reliability studies are essential for establishing the consistency of the obtained measurements of mobility, gait, and fall risk. These studies ensure that the results obtained from the TUG test or iTUG are not influenced by random variability or measurement errors, thus enhancing the validity of the assessments.

In this context, the scientific value of comparing parameters obtained from different smartphones lies in understanding the generalizability and reliability of smartphone-based assessments across various devices. This comparative analysis helps establish the consistency and robustness of the measurements, irrespective of the specific smartphone used. It also provides insights into the potential variability in sensor performance among different smartphone models, which is essential for ensuring the reliability and validity of the assessments in real-world settings.

Few investigations have studied the reliability of the iTUG considering different examiners and different tests from the same individual [6,7,8,9]. Mellone et al. (2012) [6] employed an Android smartphone and found excellent reliability between the smartphone and commercial accelerometer and intra-subject reliability between two trials recorded by the smartphone. Two additional studies [8,9] also confirmed inter-device reliability, albeit focusing solely on temporal parameters. Another limitation is that none of the studies assessed the replicability of the quantitative parameters of the iTUG across different devices.

Given the potential of smartphone inertial sensors to assess performance in the iTUG test and the existing gap in the literature, the present study aims to evaluate the reliability of the smartphone-based iTUG parameters. We proceeded with evaluations of intra and inter-device reliabilities hypothesizing that iTUG parameters are replicable in all experiments. If proven reliable, it might be a convenient and cost-effective means of assessing mobility and fall risk, potentially applied in the field of functional assessment in healthcare.

**Table 1 sensors-24-02918-t001:** Studies that have demonstrated the applicability of inertial sensors in smartphones for iTUG execution.

Autor (Year)	Smartphone/Positioning	TUG	Sensors	Gold-Standard Validation and Reliability	Objective	Population
(Silva & Sousa, 2016) [10]	Not specified/Pocket or waist or leg	30 s TUG	Acc and gyros	No	Segmentation of stages and data extraction.	Elderly individuals from the community.
(Mellone et al., 2012) [6]	HTC Desire/Lumbar	TUG extended (7 m)	Acc	Reliability	Assessing the intra-rater and inter-rater reliability of a smartphone in the TUG test.	Without criteria.
(Coni et al., 2015) [11]	Sansumg Galaxy SII and SIII/Waist	TUG	Acc and gyros	No	Classify the domains of an instrumented Timed Up and Go (TUG) test and investigate the functional decline associated with aging.	Elderly individuals.
(Galán-Mercant & Cuesta-Vargas, 2015) [12]	Iphone 4/Sternum	TUG extended (10 m)	Acc, gyros, and magnetometer	No	Determine kinematic variables that show the highest level of accuracy in discriminating between two groups of elderly individuals (frail and non-frail).	Frail and non-frail elderly individuals.
(Milosevic et al., 2013) [13]	Nexus 4/Pectoral	TUG	Acc, gyros, and magnetometer	No	Describing the parameters used to quantify the iTUG test and algorithms for extracting the parameters from signals captured by smartphone sensors.	Individuals with Parkinson’s disease and healthy individuals.
(Galán-Mercant et al., 2014) [7]	Iphone 4/Sternum	TUG extended (10 m)	Acc and gyros	Reliability	Assessing reliability and concurrent criterion validity.	Healthy older adults (5).
(Palmerini et al., 2011) [14]	HTC/Lumbar	TUG extended (7 m)	Acc	No	Identify the features that are most sensitive to locomotor performance.	Healthy individuals.
(Merchán-Baeza et al., 2018) [8]	Iphone 5s/Not specified	TUG	Acc, gyros, and magnetometer	Reliability	Collect quantitative data on lower limb function during the execution of the TUG and STS tests in individuals in the acute phase of stroke.	Elderly individuals.
(H. Yahalom et al., 2020) [15]	Iphone 5s/Sternum	TUG	Gyros	No	Characterize Parkinson’s patients and the gait performance of psychiatric patients using neuroleptic medication and compare them with Parkinson’s disease and healthy controls.	Psychiatric patients and patients with Parkinson’s disease.
(Campillay Guzmán et al., 2017) [9]	Iphone 4/Lumbar	TUG	Acc and gyros	Reliability	Assessing the reproducibility of the elapsed times between the start and end of the subphases of the TUG test.	Community-dwelling older adults.
(Ishikawa et al., 2019) [16]	Iphone/Abdomen	TUG	Acc and gyros	No	Evaluate the usefulness of measuring the times of 6 components of the TUG.	Active older adults and patients with normal pressure hydrocephalus.
(Bergquist et al., 2020) [17]	Huawei P8/Lumbar	TUG	Acc and gyros	No	Assess how well the mean features of inertial sensors could predict the total score of the CBMS.	Elderly individuals attending an outpatient clinic and healthy elderly individuals.

## 2. Methods

### 2.1. Ethical Considerations

The procedures carried out in this investigation were approved by the Research Ethics Committee of the Federal University of Pará (CAAE: 63499622.0.0000.0018), adhering to the guidelines for research involving human subjects described in Resolution 466/2012 of the National Health Council.

### 2.2. Participants

We conducted experimental procedures to evaluate intra-device reliability (Experiment A) and inter-device reliability (Experiment B). For Experiment A, a total of 30 individuals (males: 15; females: 15; mean age ± standard deviation [SD] = 26.96 ± 1.5 years; mean weight ± SD = 75.08 ± 16.03 kg; mean height ± SD = 1.69 ± 0.09 m; mean body mass index [BMI] ± SD = 26.02 ± 4.6 kg/m^2^) participated in the study, selected through convenience sampling. For Experiment B, a total of 15 individuals (males: 8; females: 7; mean age ± SD = 21.06 ± 2.76 years; mean weight ± SD = 66.8 ± 11.8 kg; mean height ± SD = 1.69 ± 0.1 m; mean BMI ± SD = 23.45 ± 3.6 kg/m^2^) made up the sample.

For both experiments, the inclusion criteria required participants to be young adults between the ages of 20 and 40, and the exclusion criteria included a history of orthopedic surgeries and musculoskeletal disorders in any limb that could be exacerbated by the test procedures. All participants completed a form providing information on their age, weight, height, and sex. No participant reported trauma in the last month. A schematic figure depicting the design of the experiments is shown in Figure 1.

### 2.3. Instruments

For Experiment A, we attached to the participant a smartphone and for Experiment B, we used three smartphones: Samsung A32, Xiaomi Redmi note 9i moto g20. Table 2 summarizes smartphone characteristics.

For both experiments, we used the Android application Momentum Science to record and save the inertial time series during the execution of the iTUG with mean sampling rate of 50 Hz. The Momentum Science App has been validated for different motion assessment protocols [18,19] and is available on the Play Store.

For all procedures, the smartphone was attached to the participant’s body using a specially designed strap and positioned at the level of the lumbar spine between the L3 and L5 vertebrae (Figure 2).

### 2.4. Experimental Protocol

For Experiment A, the subjects performed the Timed Up and Go (TUG) test three times with the same device, with a 5 min interval between each trial. The first trial was simultaneously measured using the Momentum Science App and timed using the stopwatch feature of the IWO W27 Smartwatch (Watch 7 Pro, Shenzhen, China) to compare the data related to the duration of the test. The Momentum Science App mentioned on line 172, page 6, is utilized to record and save inertial time series during the execution of the iTUG with a mean sampling rate of 50 Hz. It has been validated for different motion assessment protocols and is available on the Play Store. For Experiment B, the subjects performed three trials, but each trial used a different device. 

For all experiments, the participants were orally instructed not to use their arms for support during the rising and sitting phases. The instructions given to the participants were standardized and given by an experienced person (the same one) to facilitate understanding and minimize the risk of misinterpretation. The test path was marked with adhesive tape, including the endpoint after three meters.

The protocol followed the standard steps outlined below. The subject sat with their back in contact with the chair backrest. The data collection in the Momentum Science App was initiated by tapping the screen, and the participant remained seated for 5 s until the experimenter provided an oral command to go. In sequence, the subject stood up and walked in a straight line for three metres; after reaching a distance of three metres, the participant turned around and returned walking back to the chair. The subject turned again to sit back in the chair. Once in place, a 5 s interval was counted to allow data collection to be completed. Each participant was fitted with a smartwatch securely fastened to their non-dominant wrist. This position was chosen to minimize data variability and interference from dominant hand movements. The smartwatch was configured to ‘Flight Mode’ to prevent notifications during the test. Data collection commenced with a verbal cue, signalling the participant to start the TUG test and was manually initiated on the smartwatch. Collection ceased automatically after a predefined duration post-test completion to ensure all relevant data were captured. Time synchronization was conducted between the smartwatch and the smartphone application used for the iTUG to ensure time-stamped data from both devices were aligned for subsequent analysis. Post experiment, data from the smartwatch were downloaded via a secure Bluetooth connection to a designated computer. The time data were compared with the smartphone.

Figure 3 shows the different stages of the iTUG test.

### 2.5. Data Processing

The Momentum Science App exported the accelerometer and gyroscope recordings as text files, which were imported and analyzed using MATLAB routines (MATLAB R2015a, Mathworks, Carlsbad, CA, USA). A linear trend removal procedure was applied to the inertial time series using the detrend function. The accelerometer signals were divided by 9.81 to represent the data in gravitational units. Then, the norm of the vectors of the accelerometer and gyroscope signals was calculated according to Equation (1).
(1)$$norm=\sqrt{x^{2}+y^{2}+z^{2}}$$

On this, *x*, *y*, and *z* are the acceleration or angular velocity vectors in the mediolateral, vertical, and anteroposterior axes, respectively. 

A linear interpolation procedure was performed on the resultant vectors to achieve a sampling rate of 100 Hz since it facilitates better comparison between mobile devices. The interpolated signal was then filtered using a second-order bidirectional Butterworth filter with a cutoff frequency of 5 Hz. In summary, filtering the interpolated signal with a second-order filter having a cutoff frequency of 5 Hz is necessary to reduce noise, prevent aliasing, condition the signal, and focus on the relevant frequency content for further processing or analysis.

To identify temporal parameters and waveform amplitudes of inertial signals, the algorithm searched for 6 transient events (Figure 3):Event 1: Identification of the start of the test. The gyroscope time series was evaluated to identify the start of the rise from the chair, as previously described in the literature [20] We calculated the mean and standard deviation of the angular velocity in a time interval of 1 s in a baseline period before the rise from the chair. We considered as the onset of the test when the angular velocity exceeded this mean plus two times the standard deviation of the angular velocity in the resting time interval (red dashed line in Figure 3).Event 2: Identification of the test end. Like the identification of the movement onset, we also used the gyroscope time series to identify the test end. In this case, the time interval used as baseline was after the individual sat back in the chair. The algorithm read retrogradely the time series to indicate the end of the test as the moment which the angular velocity exceeded the mean plus two standard deviations of the resting time interval (blue dashed line in Figure 3).Events 3 and 4: Identification of the moments of the turns during the test. We observed the presence of two transient components in the angular velocity time series. The first prominent component represented the turn at 3 m of walking from the chair (here we named as G1 component in the gyroscope time series), and the second prominent component represented the turn in front of the chair to sit it back (here we named G2 component in the gyroscope time series).Event 5: Identification of the moment to complete the standing posture. To identify this event, we used the acceleration time series. We searched for the acceleration peak of a transient component just after the test onset and before the first turn moment (here we named as A1 component in the acceleration time series).Event 6: Identification of the moment when the subject sits back in the chair. Like to find the moment to complete the stand-up posture, we also used the acceleration time series to find the moment to start the transition from stand-up posture to sit back in the chair, which is represented by a transient component existing just after the second turn moment and before the test end (here we named it the A2 component in the acceleration time series).

Based on these 6 temporal markers, 11 variables of interest were calculated: total test duration, in seconds (s), representing the time interval between the test onset and test offset; go walk duration, in seconds, representing the time interval between the A1 component peak and the moment of the G1 component peak; return walk duration, in seconds, representing the time interval between the G1 component peak and G2 component peak; sit to stand duration, in seconds, representing the time interval between test onset and the A1 component peak; stand to sit duration, in seconds, representing the time interval between the A2 component and test offset; acceleration peak during the sit to standing transition, in gravitational units (g); acceleration peak during the stand to sit transition in gravitational units (g); angular velocity peak during the first turn, in radians/seconds (rad/s); angular velocity peak during the second turn, in radians/seconds (rad/s); standing jerk as the rate of acceleration change during the sit to standing transition, in g/s; and sitting jerk as the rate of acceleration change during the standing to sit transition in g/s. 

All procedures were performed automatically by the analysis routine, and a visual inspection was conducted by the researchers to detect any identification errors in the events. In case of event identification errors, visual identification was carried out by trained researchers.

### 2.6. Statistical Analysis

The statistical analysis was conducted using GraphPad PRISM 9 software. All variables were evaluated by the Shapiro–Wilk test for normality to determine the following inferential tests. Variables were considered normally distributed if *p* ≥ 0.05. For Experiment A, we compared the total duration of the test measured by stopwatch and smartphone measurements using a paired *t*-test (*t*). We also calculated Pearson’s product moment correlation (*r*) between both measurements, and we interpreted the result of the linear correlation as perfect (*r* ≥ 0.9), almost perfect (0.7 ≥ *r* < 0.9), high (0.5 ≥ *r* < 0.7), moderate (0.3 ≥ *r* < 0.5), and weak (*r* < 0.3) [21]. Furthermore, a Bland–Altman test was performed to calculate the bias and 95% limits of agreement [22]. For both Experiments, to compare the measurements obtained in the three repetitions of the iTUG test, we conducted one-way ANOVA for repeated measures, followed by Tukey’s post hoc test, if necessary. The reliability between measurements was evaluated using the intraclass correlation coefficient (ICC) for variables with normal distribution which was classified as excellent (ICC ≥ 0.75), high (0.74 ≥ ICC ≥ 0.4), and poor (ICC ≤ 0.39) [23] and Kendall’s coefficient of concordance for variables with non-normal distribution and we interpreted as excellent agreement (W ≥ 0.8), substantial agreement (0.6 ≤ W < 0.8), moderate agreement (0.4 ≤ W < 0.6), fair agreement (0.2 ≤ W < 0.4), and slight agreement (0 ≤ W < 0.2) [24]. Kendall’s coefficient of concordance is utilized when dealing with variables that do not adhere to a normal distribution, whereas the intraclass correlation coefficient is applied for variables that exhibit a normal distribution. All statistical treatments were considered significant when *p* < 0.05. For those variables which was calculated significant ICC, we additionally calculated the standard error of measurement (SEM) and minimum detectable change (MDC) using the formula:(2)$$SEM=SDpooled\times \sqrt{1-ICC}$$

*SDpooled* represents the pooled standard deviation. The MDC was calculated at a 90% level using the formula:(3)$${MDC}_{90}=SEM\times \sqrt{2}\times 1.64$$

## 3. Results

The routine written in MATLAB correctly identified the majority of repetitions (86 out of 90 records), with only four records requiring manual identification. This represents a success rate of 95.6% in the automatic identification of records and refers to the accurate identification and interpretation of the events depicted and explained in Figure 3 by the MATLAB script developed for the study. In other words, when the script correctly detected the specified events according to the predetermined criteria and definitions outlined in the research methodology, it was considered “correct”. This means that the script successfully identified the events of interest without errors or discrepancies, aligning with the expected outcomes and criteria established for the study.

### 3.1. Results of Experiment A: Comparison between Stopwatch and Smartphone Measurements

The duration of the test measured by the stopwatch and smartphone followed a normal distribution (*p* = 0.88, *p* = 0.11, respectively). Their comparison did not show significant differences (stopwatch mean test duration = 10.46 ± 1.35 s; smartphone mean test duration = 10.93 ± 1.36 s; *t*[58] = 1.315; *p* = 0.19), while the correlation between them was perfect (*r* = 0.93, *p* < 0.0001). The Bland–Altman analysis showed the agreement between devices with bias = –0.46, lower agreement limit = –1.457, upper agreement limit = 0.5348.

### 3.2. Results of Experiment A: Intra-Devicereliability of the Inertial Measurements

Table 3 shows the descriptive statistics regarding the three repetitions performed by the individuals as well as their comparison. No variable had significant differences across the sessions (*p* > 0.05).

Table 4 shows the results of the intra-device reliability evaluation for all iTUG variables. We found 9 out of 11 variables with significant intra-device reliability. Four variables had excellent reliability (test duration, stand to sit acceleration peak, return walk duration, angular velocity peak in first turn) and five variables had moderate to high reliability (go walk duration, sit to stand acceleration peak, angular velocity peak in second turn, standing jerk).

### 3.3. Experiment B Results: Inter-Device Reliability of the Inertial Measurements

Table 5 shows the descriptive statistics regarding the three repetitions performed by the individuals with different smartphone devices. Nine out of eleven variables had no significant differences across the sessions (*p* > 0.05). Post -hoc revealed that Angular velocity peak at first turn was smaller in Samsung device compared to Motorola device (*p* < 0.05), and angular velocity peak at second turn was smaller in Xiaomi device compared to Motorola device (*p* < 0.05).

Table 6 shows the results of the reliability evaluation among devices for all iTUG variables. We found 9 out of 11 variables with significant reliability, which had moderate-to-high correlation or agreement.

## 4. Discussion

The present study investigated the intra and inter-device reliabilities of features extracted from iTUG with the hypothesis that the sensors in mobile devices produce similar outcomes. Our results partially confirmed this hypothesis, as we found that most of the iTUG parameters exhibited significant intra and inter-device reliability. The study yielded several key findings regarding the performance and reliability of inertial measurements in the context of the Timed Up and Go (iTUG) test, as well as the comparison between stopwatch and smartphone measurements. Firstly, the MATLAB routine demonstrated a high success rate of 95.6% in automatically identifying trial records, with only a small percentage requiring manual intervention. In Experiment A, the comparison between stopwatch and smartphone measurements revealed no significant differences in test duration, with both methods displaying a normal distribution and a strong correlation. Bland–Altman analysis confirmed agreement between devices. Additionally, intra-device reliability analysis showed no significant differences in iTUG variables across sessions, with most variables demonstrating significant intra-device reliability. Notably, nine out of eleven variables exhibited significant reliability, with four variables showing excellent reliability and five displaying moderate to high reliability.

Notably, nine out of eleven variables exhibited significant reliability, with four variables showing excellent reliability, and five displaying moderate to high reliability. In Experiment B, inter-device reliability analysis indicated no significant differences for the majority of variables across sessions, although some discrepancies were observed between specific smartphone models in angular velocity peak at the first and second turn. Overall, the study underscores the potential of inertial measurements for reliable and accurate assessment in mobility-related tests, with implications for both research and clinical practice.

The present study demonstrated the accuracy of the iTUG tool for automatically measuring the test completion time, as there was an excellent correlation between the measurements obtained by a human using a stopwatch and those obtained with smartphones. These findings align with the previous literature, although they have used portable accelerometers and not smartphones, demonstrating consistent results across various subject groups, including healthy control, individuals with Parkinson’s disease [25], and the elderly [26]. These previous studies also observed excellent correlations in these diverse cohorts, indicating the reliability and validity of our study outcomes. This consistency across different populations strengthens the robustness of our findings and underscores the potential applicability of our research in broader contexts.

Regarding applicability, the test was conducted on thirty subjects without repetition and without data loss from the sensors during the assessments. The routine written in MATLAB correctly identified the majority of repetitions (86 out of 90 records), with only four records requiring manual identification. This represents a success rate of 95.6% in the automatic identification of records. Thus, it can be inferred that the use of smartphone inertial sensors recorded by the Momentum application can be performed by other individuals without any loss in the quality of the collected data. Moreover, this applicability adds to the future possibility of self-application of the iTUG. This applicability confirms what has already been demonstrated in previous studies involving the use of smartphone inertial sensors, which have shown the application of the device in the evaluation of additional TUG parameters [7,8,9,13,14,15,16,27,28]. In this study, we explored the significance of calculating the norm (magnitude) of acceleration and angular velocity data series in three axes, shedding light on its importance in motion analysis. Our findings underscore the utility of this approach, as it provides a scalar representation of the overall magnitude of motion, simplifying analysis and interpretation. Moreover, the norm offers a practical means of reducing the dimensionality of data, facilitating comparisons across different contexts while remaining invariant to sensor orientation. Despite these advantages, it is noteworthy that few studies have delved into the analysis of the norm, highlighting a gap in the existing literature [27]. Our study stands out as the first to rigorously examine the replicability of the norm, thereby contributing a novel perspective to the field. By addressing this underexplored aspect, we have expanded our understanding of motion analysis and provided valuable insights that pave the way for future research endeavors in this area.

In the intra-device evaluation, the parameters exhibited no significant differences across attempts, with nine of them demonstrating replicability. The substantial proportion of replicable parameters underscores the reliability of the trials. However, it remains unclear why the variables ‘Sit to Stand Duration’ and ‘Stand to Sit Duration’ did not exhibit significant replicability. In particular, these two variables, characterized by their short duration, may be susceptible to influences arising from the participant’s immobility at the test’s initiation and conclusion. The automatic detection of test onset and completion times is derived from gyroscope signals, and minor fluctuations related to trunk movement during these instances can introduce noise, potentially affecting the precision of detection. Given the brevity of these variables, slight variations in detecting the test’s start and end may have contributed to their increased variability across the three attempts.

In the context of automatic detection, the term “algorithm” refers to a set of predefined rules or instructions implemented within a given script to perform specific tasks related to event detection or data processing. The algorithms for detecting phases in the Timed Up and Go (TUG) test vary in the literature. For example, Silva & Sousa (2016) [10] manually and automatically segmented the phases, with the former being carried out through comparison with a video recording and the latter based on the integral of the gyroscope signal to identify turning points, defining transitions between sitting and standing postures when consecutive differences of 3 degrees in the angle signal occurred. In our study, we used the angular velocity signals from the gyroscope to define the start and end of the test, as well as the peaks in the accelerometer signal to identify posture transitions, reducing the risk of an unreliable analysis through image observation. In th study by Coni et al. (2015) [11], similar to ours, the accelerometer and gyroscope were used, and although the identification of turns was described based on the angular velocity signal, similar to what was used in the present study, it was not specified which signals were used to define the start and end of the test. Ishikawa et al. (2019) [16] also identified test phases based on gyroscope signals, such as the time taken to rise from the chair at the beginning of the test; however, the obtained variables, unlike in our study, were only related to the time of each phase.

In the inter-device comparisons, it was observed that the maximum angular velocity at both turning moments exhibited higher values on the Motorola device compared to one of the other devices. Although not the focus of this study, the literature suggests evidence that the sensor’s weight may impact measurement sensitivity [28]. Two parameters (i.e., ‘Stand to Sit Duration’ and ‘Sitting Jerk’) showed no significant inter-device reliability. ‘Stand to Sit Duration’ continues to lack significant replicability, potentially influencing the absence of replicability in the ‘Sitting Jerk’ parameter.

Previous studies have also investigated the reliability of smartphone sensors, such as Mellone et al. (2012) [6], where reliability ranged from fair to excellent in almost all parameters, both intra- and inter-rater evaluations. Their results, like ours, showed that the total duration of the test is a variable with excellent reliability, and the rise jerk is a variable with moderate to high reliability. In the study by Galán-Merchant et al. (2014) [7], reliability among older adults was tested, and the findings indicate excellent reproducibility in acceleration magnitude and displacements along the vertical, mediolateral, and anteroposterior axes. Nevertheless, the study’s sample size was limited to just five participants. While the test phases followed established protocols from prior publications, the analysis solely focused on variables associated with acceleration in the motion axes, such as peak values and acceleration magnitude. Furthermore, unlike our study, the investigation conducted an extended version of the test, encompassing a 10 m walking distance, and positioned the smartphone towards the sternum. Other authors [7,8] also tested the reliability of smartphone inertial sensors and obtained excellent reliability results for all variables. However, only time variables for each stage of the test were compared.

In acknowledging the limitations of our study, it is important to note that while the smartphone-based measurements employed in the iTUG experiment provided valuable insights, the validity of our method could be significantly enhanced by a comparative analysis with data obtained from an inertial measurement unit (IMU). Furthermore, the study’s limitations encompass the lack of validation between smartphone inertial sensors and kinematics, especially regarding variables beyond the TUG execution time. Additionally, certain variables exhibited limited replicability, indicating potential areas for refining the test protocol or analysis methods. Moreover, forthcoming research could leverage alternative smartphones to assess data reliability employing diverse tools. Furthermore, we recognize the study’s limited generalizability to other age groups and advocate for exploring its effects across broader age ranges.

This research confirmed the validity, applicability, and reliability of smartphones in measuring additional parameters to the Timed Up and Go test, demonstrating the possibility of using low-cost devices for evaluating important variables related to individuals’ mobility and balance. The use of these sensors in conjunction with the Momentum application allows for accurate and automated identification of events and calculation of variables, enabling the assessment of additional parameters and the potential for self-application. These results contribute to the growing body of evidence supporting the use of smartphone sensors as a low-cost and feasible option for clinical assessments in various contexts.

## Figures and Tables

**Figure 1 sensors-24-02918-f001:**
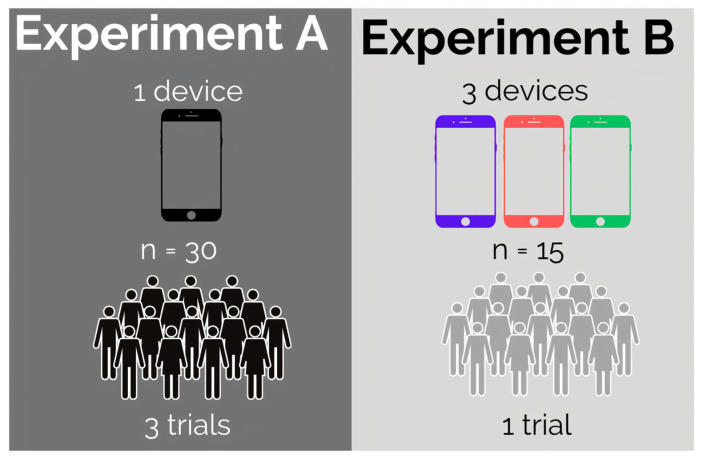
Summary of the experiment’s design.

**Figure 2 sensors-24-02918-f002:**
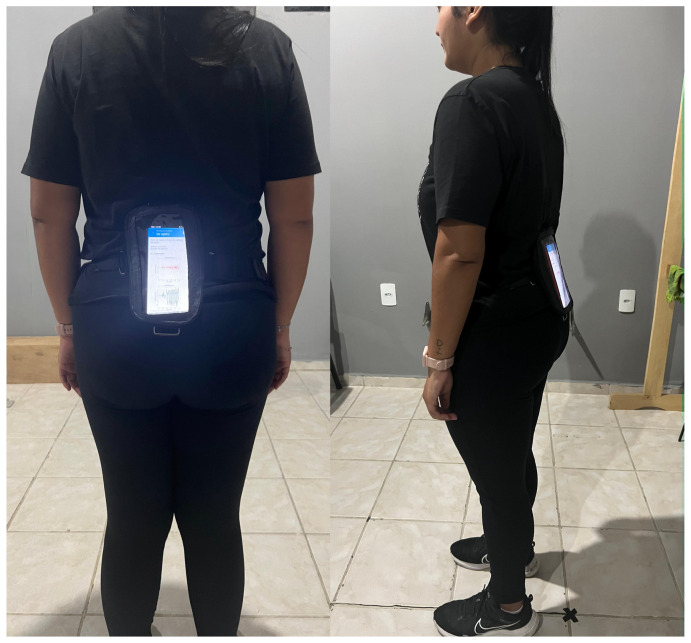
Smartphone attachment.

**Figure 3 sensors-24-02918-f003:**
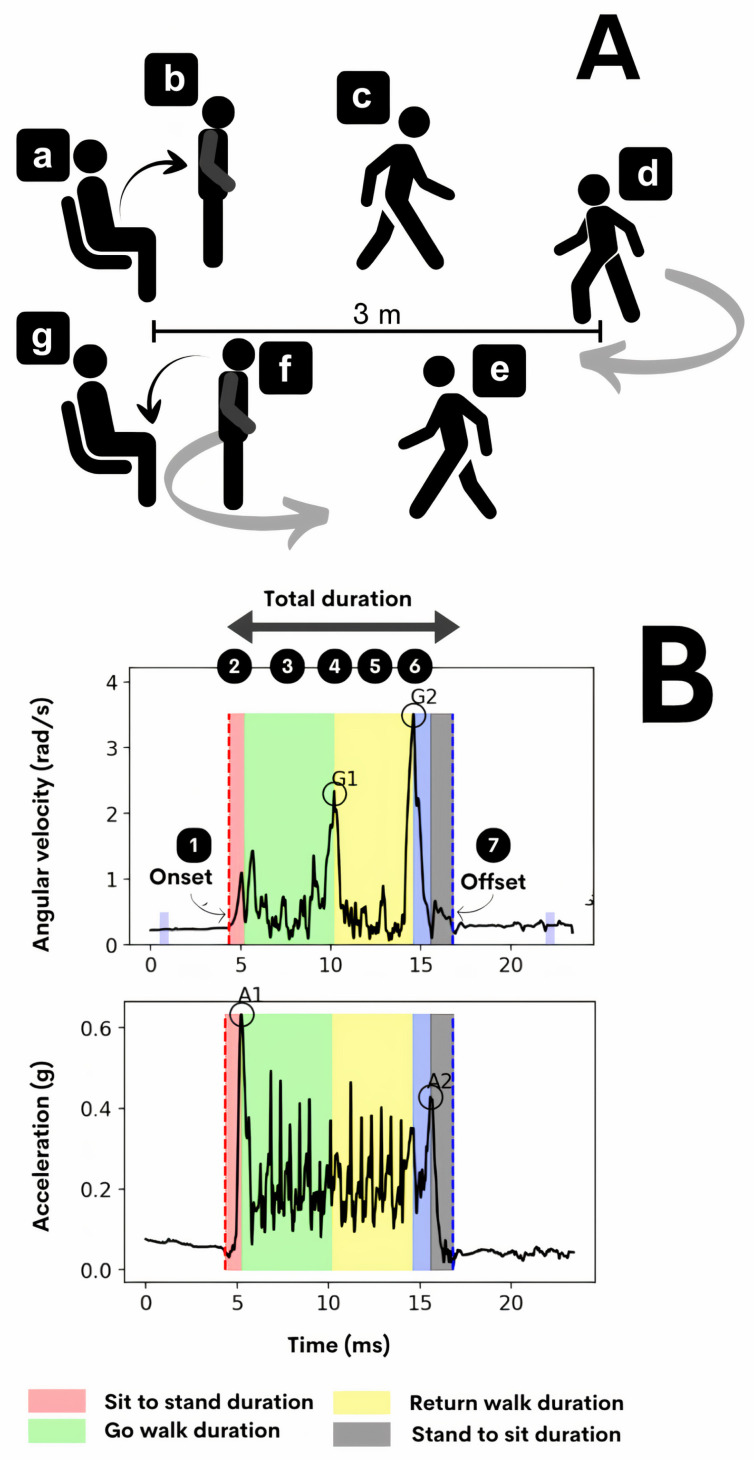
TUG test phases (**A**) and smartphone-based iTUG assessment (**B**). In (**A**), the phases involved rising from a chair (a,b), walking along a 3 m path (c), making a turn to return to the chair (d), walking back to the chair (e), making a second turn to sit in the chair (f) and sitting down (g). Inertial time series data were collected during the test and subsequently analyzed to derive performance-related features. In (**B**), the accelerometer and gyroscope signals during the test are represented, as well as the segmentation of phases as explained below (events 1 to 7). The signals depicted in this figure represent the norms.

**Table 2 sensors-24-02918-t002:** Smartphone’s characteristics.

Device	Dimensions	Weight	Inertial Unit	Gyroscope Range/Resolution	Accelerometer Range/Resolution
Xiaomi Redmi Note 8	158.3 × 75.3 × 8.4 mm,	190 g	Mode BOSCH	±2000 degrees per second/16-bits	±16 g 16 bit
Samsung A32	158.9 × 73.6 × 8.4 mm	184 g	model LSM6DSL	±2000 degrees per second/16-bits	±16 g 16 bit
Xiaomi Redmi note 9i	164.9 × 77.1 × 9 mm	194 g	model: bmi260	±2000 degrees per second/16-bits	±16 g 16 bit
moto g20	165.2 × 75.7 × 9.2 mm	200 g	icm40607	±2000 degrees per second/16-bits	±16 g 16 bit

**Table 3 sensors-24-02918-t003:** Descriptive statistics and comparison of the iTUG variables across the sessions (experiment A). The values are represented by median (interquartile range) or mean ± standard deviation regarding the data distribution adjustment to the gaussian fit. *p*-value refers to one-way ANOVA result.

Variable	Session #1	Session #2	Session #3	*p*-Value
Test duration (s)	10.71 (1.73)	10.15 (1.87)	10.24 (1.87)	0.47
Sit to stand duration (s)	0.98 (0.02)	0.8 (0.02)	0.98 (0.02)	0.61
Go walk duration (s)	4.33 (0.51)	4.26 (0.64)	4.14 (0.64)	0.46
Return walk duration (s)	3.78 (0.95)	3.66 (1.08)	3.55 (1.15)	0.92
Stand to sit duration (s)	1.46 (0.37)	1.32 (0.43)	1.4 (0.57)	0.07
Sit to stand acceleration peak (g)	0.57 ± 0.11	0.57 ± 0.11	0.58 ± 0.11	0.51
Stand to sit acceleration peak (g)	0.54 (0.17)	0.52 (0.09)	0.54 (0.15)	0.8
Angular velocity peak in first turn (rad/s)	3.54 ± 0.81	3.64 ± 0.9	3.57 ± 0.86	0.42
Angular velocity peak in second turn (rad/s)	3.83 ± 0.7	3.9 ± 0.73	3.94 ± 0.86	0.57
Standing jerk (g/s)	0.58 ± 0.13	0.59 ± 0.12	0.59 ± 0.11	0.91
Sitting jerk (g/s)	0.38 (0.15)	0.4 (0.16)	0.39 (0.17)	0.45

**Table 4 sensors-24-02918-t004:** Intra-device reliability (experiment A) of the iTUG variables. For normally distributed variables, we report the intraclass correlation coefficient (ICC); otherwise, Kendall’s W agreement is reported.

Variable	Reliability	95% CI	*p*-Value	SEM	MDC
Test duration (s)	W = 0.89	0.78–0.93	<0.0001	-	-
Sit to stand duration (s)	W = 0.2	–0.56–0.4	0.96	-	-
Go walk duration (s)	W = 0.79	0.57–0.85	<0.001	-	-
Return walk duration (s)	W = 0.82	0.66–0.87	<0.001	-	-
Stand to sit duration (s)	W = 0.42	–0.1–0.55	0.15	-	-
Sit to stand acceleration peak (g)	ICC = 0.73	0.57–0.85	<0.0001	0.29	0.81
Stand to sit acceleration peak (g)	W = 0.85	0.71–0.89	<0.001	-	-
Angular velocity peak in first turn (rad/s)	ICC = 0.86	0.76–0.93	<0.0001	0.59	1.64
Angular velocity peak in second turn (rad/s)	ICC = 0.73	0.57–0.85	<0.0001	0.77	2.14
Standing jerk (g/s)	ICC = 0.59	0.39–0.76	<0.0001	0.37	1.03
Sitting jerk (g/s)	W = 0.58	0.18–0.71	0.008	-	-

W: Kendall’s W agreement; ICC: intraclass correlation; CI: confidence interval.

**Table 5 sensors-24-02918-t005:** Descriptive statistics and comparison of the iTUG variables among devices (experiment B). The values are represented by media (interquartile range) or mean ± standard deviation regarding the data distribution adjustment to the gaussian fit. *p*-value refers to one-way ANOVA result.

Variable	Samsung	Xiaomi	Motorola	*p*-Value
Test duration (s)	11.82 ± 0.94	11.83 ± 0.96	11.75 ± 1.03	0.93
Sit to stand duration (s)	1 (0.02)	1 (0.16)	1 (0.02)	0.23
Go walk duration (s)	4.41 ± 0.48	4.3 ± 0.62	4.17 ± 0.53	0.18
Return walk duration (s)	3.86 (1.06)	3.79 (1.01)	4.06 (1.18)	0.5
Stand to sit duration (s)	1.52 (0.56)	1.75 (0.67)	1.41 (0.52)	0.1
Sit to stand acceleration peak (g)	0.52 (0.17)	0.5 (0.17)	0.49 (0.18)	0.61
Stand to sit acceleration peak (g)	0.51 ± 0.15	0.54 ± 0.15	0.52 ± 0.13	0.68
Angular velocity peak in first turn (rad/s)	2.46 (0.53)	2.46 (0.31)	2.63 (0.36)	0.01
Angular velocity peak in second turn (rad/s)	2.92 (0.66)	2.85 (0.16)	2.92 (0.78)	0.032
Standing jerk (g/s)	0.52 (0.17)	0.47 (0.22)	0.46 (0.14)	0.79
Sitting jerk (g/s)	0.34 ± 0.14	0.31 ± 0.12	0.37 ± 0.13	0.46

**Table 6 sensors-24-02918-t006:** Inter device reliability (experiment B) of the iTUG variables. For normally distributed variables, we report the intraclass correlation coefficient (ICC); otherwise, Kendall’s W agreement is reported.

Variable	Reliability	95% CI	*p*-Value	SEM	MDC
Test duration (s)	ICC = 0.56	0.25–0.81	0.0001	1.1	3.04
Sit to stand duration (s)	W = 0.6	0.23–0.72	0.03	-	-
Go walk duration (s)	ICC = 0.55	0.26–0.8	0.0001	0.83	2.29
Return walk duration (s)	W = 0.83	0.69–0.87	0.001	-	-
Stand to sit duration (s)	W = 0.35	–0.26–0.5	0.41	-	-
Sit to stand acceleration peak (g)	W = 0.6	0.22–0.71	0.03	-	-
Stand to sit acceleration peak (g)	ICC = 0.56	0.25–0.8	0.0001	0.42	0.17
Angular velocity peak in first turn (rad/s)	W = 0.75	0.54–0.78	0.005	-	-
Angular velocity peak in second turn (rad/s)	W = 0.74	0.52–0.8	0.005	-	-
Standing jerk (g/s)	W = 0.71	0.46–0.78	0.008	-	-
Sitting jerk (g/s)	ICC = 0.02	–0.23–0.4	0.42	-	-

W: Kendall’s W concordance; ICC: intraclass correlation; CI: confidence interval.

## Data Availability

Supplementary data available following MDPI Research Data Policies.

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
