# Peer review of "Intra and Inter-Device Reliabilities of the Instrumented Timed-Up and Go Test Using Smartphones in Young Adult Population"

_sensors, 2024, doi:10.3390/s24092918_

Round 1
Reviewer 1 Report
Comments and Suggestions for Authors
The paper presents a set of experiments to detect the phases of the TUG movement using smartphones. The paper is very well written and clearly explained. However I consider that it can be improved in the next terms:
- The abstract is too long. Indeed, the abstract shows the results obtained even with statistical values. The abstract should be shortened and give general information about the experiments and final results but not with values but with text summarizing the results.
- In section "Instruments" a table summarizing the smartphones characteristics would allow an easy comparison between them.
- Figure 2 has not a description. The description is "Insert here"
- The samplig rate used is 50 Hz. Afterwards, it is augmented to 100 Hz using interpolation. Why? Is it really necessary? If so, explain why
- The interpolated signal is filtered using a second-order filter with cutoff frequency of 5Hz. Explain why it is required
- The event numbering is confusing as it is related to Figure 2 that uses a different numeration. Event 1 is not the number 1 of the figure. Maybe it would be better to use letters instead of numbers.
- In page 8 all the ranges are shown from lower to higher calues (W for instance). But ICC is shown from higher to lower values. It would be better to maintain the same scheme for all of them (from low to high)
- The text "(Equation 2)" and "(Equation 3) should not be so close to the equation. Move them to the right.
- The description of tables 4 and 5 are exactly the same although they present data from different experiments. Differentiate them.
- Try not to break Table 5 into two pages
Reviewer 2 Report
Comments and Suggestions for Authors
The paper explores the reliability of the Instrumented Timed-Up and Go (iTUG) test using smartphones for assessing mobility in young adults. It conducts intra-device and inter-device reliability assessments, hypothesizing that iTUG parameters are consistent across all experiments. The study finds that most iTUG parameters showed significant reliability, indicating smartphones with inertial sensors are a viable option for clinical TUG assessments. However, variations in angular velocity peaks among devices suggest the need for consideration of device differences in measurements.
- The footnotes for each figure must contain clear content (e.g., "Figure 1. About here"), and the text and diagrams within the figures must be clearly visible.
- Since the graph in Figure 2 contains more data than the left side's experimental situation, it needs to be presented in a way that makes it clearly visible.
- The overall format of the paper differs from that of the "Sensors" journal. Please check and rewrite accordingly.
- It seems that the discussion part includes a comparison of the results of this paper with related studies, including recent research. The contents of this study need to be synthesized to write an additional Conclusion part.
- This paper has limited its experimental subjects to young adults. Limited to a young adult population, raising questions about applicability to other age groups. What is the reason for limiting it to young adults?
- There might be device-specific differences in certain iTUG parameters, is there no problem with standardization among various smartphone models? It depends on a specific app, but is it universally usable on all platforms or devices? Additional research is needed to validate and standardize the iTUG application across various smartphone devices and different experimental groups.
- Some variables did not show significant replicability, suggesting possible improvements in the test protocol or analysis methods.
Comments on the Quality of English LanguageThere doesn't seem to be any major problem with grammar. Please reconsider dividing the paragraphs too briefly, and review the paper as a whole.
Reviewer 3 Report
Comments and Suggestions for Authors
Thanks for offering the opportunity to review this piece of work. The data collection part of this work is substantial. However, I have concerns about the novelty and contribution of the work. My detailed comments are below:
Introduction:
Please elaborate more on what 'gold standard' validation involves.
The contribution and novelty of the work need to be clearly stated.
The scientific value of comparing parameters obtained from different smartphones is not clear.
Method
Figure 1: what does the circular symbol (at the bottom of the figure) mean?
What is the justification for choosing the Xiaomi phone as the device for experiment A?
Units, such as 'dps' needs to be introduced.
To aid clearer comparison, it is recommended to transform text between L134 and 141 into a table.
It would be helpful to have a photo accompanying text between L146 and L148 showing the setup.
What is the reason for placing the phone between L3 and L5?
Authors need to clarify the definition of a test and a trial.
what does the linear trend removal procedure involve?
What is the purpose of calculating the norm of the vectors for both acceleration and angular velocity?
What is the purpose of up-sampling the data to 100Hz, which is typically not recommended?
The word 'algorithm' is introduced, but never defined what the algorithm involves.
L225: it is not clear why jerk was chosen to be a parameter of interest.
Results
L258: what is the definition of 'correct'.
L263: the threshold of the p-value for the normal test needs to be explained in the method section.
L265, t[58]=1.315, t is never introduced.
Table 3: it is unclear why W and ICC were used randomly in the table. In addition, what the negative value for duration entail? what does the division symbol mean, as part of the 95% confidence interval?
The p value presented in table 2 and table 4, does mean the significance in difference among all three groups or two particular groups?
Discussion
L369-L370, it is not clear what do the authors mean by 'this study', but 'unlike ours'.
L355, what are the two parameters?
L356-L357: what does it mean by replicability?
The contribution of this work is not clearly stated in the discussion.
Comments on the Quality of English Language
English is poor and requires extensive editing.
Round 2
Reviewer 3 Report
Comments and Suggestions for Authors
Thanks for revising the manuscript. I can confirm that my concerns have been addressed.
Comments on the Quality of English LanguageEnglish edits are still needed for the manuscript.
Author Response
English was revised